# Pattern Learning and Knowledge Distillation for Single-Cell Data Annotation

**DOI:** 10.3390/biology15010002

**Published:** 2025-12-19

**Authors:** Ming Zhang, Boran Ren, Xuedong Li

**Affiliations:** 1Alibaba Business School, Hangzhou Normal University, Hangzhou 311121, China; 2Luoyang Central Hospital, Zhengzhou University, Luoyang 471000, China; 3School of Intelligent Systems Engineering, Shenzhen Campus, Sun Yat-Sen University, Shenzhen 518107, China; renbr@mail2.sysu.edu.cn; 4College of Artificial Intelligence (CUIT Shuangliu Industrial College), Chengdu University of Information Technology, Chengdu 610225, China; xuedongl@cuit.edu.cn

**Keywords:** pattern learning, knowledge distillation, cell type annotation, batch integration

## Abstract

Single-cell technologies allow researchers to measure gene expression at the level of individual cells, providing a powerful way to study cell identities and understand biological processes. However, differences between datasets—known as batch effects—can make accurate cell type annotation challenging. The existing deep learning methods often require heavy computation or fail to jointly address batch correction and cell type identification. In this study, we introduce PLKD, a new method that uses biologically meaningful gene groups, called patterns, together with knowledge distillation to improve cell type annotation. PLKD first learns pattern-level information using a Transformer-based Teacher model and then transfers this knowledge to a lightweight Student model. This design enables PLKD to accurately classify cell types, reduce batch effects, and provide interpretable biological insights. Our results show that PLKD achieves high accuracy and robustness while remaining efficient for large-scale datasets, offering a practical and interpretable tool for single-cell analysis.

## 1. Introduction

Single-cell measurement technology can measure the abundance of molecules at the cellular level. Single-cell data (such as the most common scRNA-seq) analysis allows us to understand the life status of organisms more clearly and screen drugs for diseases [1,2,3,4,5]. In AI-based single-cell data analysis, cell type annotation (or classification) based on the gene expression values of cells is the most basic issue [6]. Aiming at the problem of cell type annotation, many basic framework pipelines [7] have been developed, such as Seurat [8], scanpy [9] and scverse [10]. These classic methods usually include preprocessing, dimensionality reduction, clustering, differential analysis, and manual annotation based on prior knowledge. Therefore, these commonly used software packages rely on feature selection and linear dimensionality reduction before clustering single-cell data, which may result in information loss. In contrast, deep learning methods provide a promising solution for modeling non-linear relationships among all genes [11].

Currently, cell type annotation methods based on deep learning are divided into two categories: supervised and unsupervised. Benchmark studies have shown that supervised cell type annotation methods have more advantages in terms of accuracy and robustness [12,13]. In addition, supervised methods fit the actual application scenario: bioinformatics researchers manually annotate cell types on the batch data of the first few measurements, then design a model to learn the labeled data, and use the model to annotate the batch data obtained in the future (no need to re-training). Therefore, transferring cell type annotations from the reference dataset to the new query dataset remains a valuable problem. For recent supervised methods, scGPT [14] and scBERT [15] use large language models as the foundation, which require expensive computing power. Although Cellcano [16] and TOSICA [17] do not need high computational complexity, the accuracy of classification results is lower than scGPT and scBERT. In addition, compared with the fine-tuning strategies of scBERT and scGPT, TOSICA and Cellcano require a lot of epochs to train from scratch. Recently, KIDA [18] employed knowledge distillation and demonstrated that it can improve the robustness of test results. However, KIDA’s process is complex and requires improvement.

Single-cell measurement techniques inevitably lead to domain gaps in multiple batches or datasets, that is, the batch effects. The existing supervised methods almost do not emphasize the removal of batch effects in the latent space of cells. For the multi-source data, batch effects can interfere with cell type discriminability [19,20]. For batch effects removal, popular methods such as MNN [21] rely on mutual nearest neighbor methods to eliminate batch effects. Such methods require more computational overhead. For example, MNN can only analyze two batches at a time. When the number of batches increases, it can easily lead to computational infeasibility. According to benchmark studies [22,23], in terms of running time, Harmony [24] can be used as the first attempt. The current standard approach is to first use these batch correction methods to remove batch effects to obtain a joint embedding of multiple batches of data, and then use the cell embedding for cell type identification. Under such a process, the cell typing task and the batch integration task are considered independently. In contrast, multi-task learning [25,26,27] has proven its effectiveness in biological data analysis [28,29,30,31]. UnitedNet simultaneously learns the two tasks of multi-modal integration and cross-modal prediction. Ablation experiments of UnitedNet show that the multi-task learning method achieves better results than learning two tasks independently [32].

For single-cell data, batch correction should not only eliminate gaps between batches, but also improve the biological heterogeneity of clusters, which is equivalent to improving the discriminability of cell representations [33,34,35]. Therefore, batch correction and cell typing are two tasks, which closely linked and reinforce each other. On the one hand, we can achieve batch correction through a clustering-based loss function [36,37,38], on the other hand, we can design a specific network architecture. We are inspired by the pathway-based method [17]. We group all genes into different gene sets (patterns), and each pattern represents a specific function. This design enables model to focus on biologically relevant functions interaction rather than gene-level expression that is susceptible to batch effects. Intuitively, this design can be illustrated by image recognition [39,40]: although the animals are photographed from different angles, species experts can still identify subtypes from the photos. The relationship between patterns and the semantics of patterns both are discriminative information that is not affected by the camera angle, as shown in Figure 1a. For single-cell data, batch effects correspond to pixel differences caused by camera angle. The pattern is a collection of genes, representing a function. Furthermore, patterns are interpretable. Enrichment analysis can annotate the pattern as a pathway, see Figure 1b.

Based on the above motivations, we propose PLKD, a single-cell data annotation method based on pattern learning and knowledge distillation. PLKD comprises a Teacher model (Transformer) and a Student model (MLP). The Teacher model learns patterns from the reference dataset using a self-attention mechanism, and then transfers these patterns to the lightweight Student model through knowledge distillation. The Teacher model learns batch integration capabilities from the data, while the Student model acquires knowledge from the Teacher model through knowledge distillation, demonstrating robust performance. Furthermore, the simple MLP simplifies cell annotation on large-scale query datasets. By training on the reference dataset, PLKD can eliminate batch effects in both the query and reference datasets and identify clusters with biological heterogeneity, enabling bioinformatics researchers to more deeply differentiate the cellular characteristics and functions of different biological systems. Based on the Student MLP architecture, and combined with SHAP [41], we can extract interpretable information from PLKD, i.e., tracing back to gene-level inputs based on cell typing results. Therefore, we can infer potential marker genes for specific cell types. We built PLKD based on the TOSICA framework [17]. Benchmark experiments demonstrate that PLKD enables accurate and robust cell type annotation, batch integration, and even multi-modal integration.

## 2. Materials and Methods

### 2.1. The Overall of PLKD

PLKD contains Teacher and Student. The former learns patterns (feature subsets with biological function) from reference dataset based on the self-attention, and the latter transfers Teacher’s knowledge to lightweight Student through distillation technology. The inference and training settings of PLKD are shown in Figure 2a. Note that the current discussion applies to RNA modality data (abbreviated as scRNA-seq), which is characterized by gene expressions. In Teacher, the Navigator module converts single-cell from input gene expressions into *m* patterns. Additional ‘cls token’ is added to the input of the self-attention layer as the representation of the current cell. The Classifier is used to assign cells to their groups (categories). KL loss transfers the knowledge of Teacher to Student. Divergence-based clustering loss and self-entropy loss are used to enhance the discriminability of cell representations output from Teacher and Student, respectively, while assisting batch integration. When labels are available in reference dataset, cross-entropy loss can further improve the discriminability of cell representations.

### 2.2. Inference

PLKD uses the Navigator module to convert gene expressions into pattern tokens. The Navigator module is shown in Figure 2b. For cell *i*, it contains *n* genes. The gene expressions of this cell is $x_{i}\in R^{n}$. Navigator module uses a set of linear matrices $\left\{W_{j}\in R^{n\times h}|j=1,\ldots ,m\right\}$ to process $x_{i}$ in parallel and convert the cell into *m* patterns $\left\{v_{j}\in R^{h}|j=1,\ldots ,m\right\}$. Each linear matrix represents a pathway pattern. Navigator module contains an optional step where we can add knowledge-based masks, which can limit the representation of the pattern during the model initialization. For the knowledge-based mask, we create a list consisting of all gene names in reference dataset, and obtain *m* pathways corresponding to this gene list through GSEA (Gene Set Enrichment Analysis) combined with pathway annotation, see Figure 2c. We added mask to the linear matrix according to the relationship between genes and pathways, see Figure 2d. Assuming that gene 1 is not in pathway 1, the first row (gene 1) of linear matrix $W_{1}\in R^{n\times h}$ (pathway 1) is all masked to value 0. On the contrary, assuming that gene *n* is in pathway 1, the *n*-th row of linear matrix $W_{1}\in R^{n\times h}$ will not have a mask added.

Typically, the number of patterns compared to the number of genes is m<<n, so when we use standard self-attention layer [42] to process these patterns, it generally does not cause a large overhead in computational resources. First, we add a ‘cls token’ $c\in R^{h}$ (a learnable parameter) to represent the global information of current cell. We concat all tokens as $I=concat\left(v_{1},\ldots ,v_{m},c\right)\in R^{h\times (m+1)}$. The query, key and value are $Q=W^{Q}I$, $K=W^{K}I$ and $V=W^{V}I$, respectively. The self-attention can be defined as(1)$$\begin{array}{c} Attn=Attn\left(Q,K,V\right)=softmax\left(\frac{QK^{⊤}}{\sqrt{h}}\right)V. \end{array}$$

In PLKD, the self-attention layer has only 1 layer, including 2 heads. We have(2)$$\begin{array}{c} O=W^{O}concat\left(head_{1},head_{2}\right), \end{array}$$
where $head_{i}$ is(3)$$\begin{array}{c} head_{i}=Attn\left(W_{i}^{Q}Q,W_{i}^{K}K,W_{i}^{V}V\right). \end{array}$$

The self-attention layer outputs more discriminative updated pattern tokens $\left\{r_{j}\in R^{h}\right\}$ (j=1,…,m) and updated cls token $c_{r}\in R^{h}$. We use $c_{r}$ as the embedding of the cell. The output contains(4)$$\begin{array}{c} O=concat(r_{1},\ldots ,r_{m},c_{r}). \end{array}$$

With the $c_{r}$ of all cells, UMAP coordinates are obtained through *scanpy.tl.umap* to visualize all cells’ embedding. For the type annotation of cell *i*, we use a fully connected network as the classifier. We map the embedding $c_{r}$ of cell *i* to the corresponding category, and record the classification logits output (pre-prediction probability) by the classifier as $y_{i}^{teacher}\in R^{k}$, where *k* represents the total number of category labels. In PLKD, Student is set up as a simple MLP (Multi-Layer Perceptron). The input of Student is $x_{i}$ and the output logits (final prediction probability) is $y_{i}^{student}\in R^{k}$.

### 2.3. Training

In inference stage of PLKD, Navigator uses a naive but special network to obtain patterns. In order to ensure that PLKD correctly learns the discriminability of each cell, we need additional goals to guide Teacher and Student.

For Teacher, we set two loss functions: divergence-based clustering loss and cross-entropy loss. They allow the cell embedding to be separated between classes and compact within classes. Divergence-based clustering loss reduces the correlation between the predicted logits (soft labels) of different types of clusters, thereby increasing the discriminability between clusters. We represent correlation by the similarity between clusters. Let the number of cells in reference be *N* and define a similarity matrix $S\in R^{N\times N}$, element $s_{i,j}=exp(-||c_{r,i}-c_{r,j}{\left|\right|}^{2})$ represents the similarity between two cells, $c_{r,i}$ and $c_{r,j}$ represent the embedding of cell *i* and cell *j* in reference, respectively. Define $Y_{a}^{teacher}=\left[y_{1}^{teacher}\left[a\right],\ldots ,y_{N}^{teacher}\left[a\right]\right]$ as the logits of cells 1 to *N* classified into category *a*, where $y_{1}^{teacher}\left[a\right]$ is the *a*-th element of $y_{i}^{teacher}$. Minimizing $Y_{a}^{teacher}S{\left(Y_{b}^{teacher}\right)}^{⊤}$ means that we want to make $Y_{a}^{teacher}$ and $Y_{b}^{teacher}$ orthogonal. Therefore, divergence-based clustering loss is(5)$$\begin{array}{c} L_{1}=\\ \frac{1}{k}\sum_{a=1}^{k-1}\sum_{b>a}^{k}\frac{Y_{a}^{teacher}S{\left(Y_{b}^{teacher}\right)}^{⊤}}{\sqrt{Y_{a}^{teacher}S{\left(Y_{a}^{teacher}\right)}^{⊤}Y_{b}^{teacher}S{\left(Y_{b}^{teacher}\right)}^{⊤}}}. \end{array}$$

When labels in reference are available, cross-entropy loss helps divergence-based clustering loss converge quickly. Let the one-hot label of the *i*-th cell be $g_{i}$, and cross-entropy loss is(6)$$\begin{array}{c} L_{2}=-\frac{1}{N}\sum_{i=1}^{N}g_{i}^{⊤}log\left(softmax\left(y_{i}^{teacher}\right)\right). \end{array}$$

In order to accelerate large-scale query dataset cell type annotation, we obtain a lightweight Student based on Teacher. Knowledge distillation transfers the knowledge learned from Teacher to the lightweight Student. Knowledge distillation usually uses the prediction results produced by the Teacher model as soft labels for training the Student model [43]. In PLKD, knowledge distillation can alleviate the label noise problem. Specifically, the training target of Student comes from the soft labels output by Teacher. Compared with hard labels that only contain 1 and 0, soft labels are smooth and therefore more tolerant of label noise in the dataset. Therefore, knowledge distillation can work in scenarios where annotations are incorrect. Let KL(P,Q) be the *KL* divergence of two probability distributions *P* and *Q*, T=3 be the temperature parameter, and the loss function of knowledge distillation is(7)$$\begin{array}{c} L_{3}=\\ \frac{1}{N}\sum_{i=1}^{N}KL\left(log\left(softmax\left(\frac{y_{i}^{student}}{T}\right)\right),softmax\left(\frac{y_{i}^{teacher}}{T}\right)\right). \end{array}$$

Knowledge distillation alleviates noise problem, but soft labels cause the logits output by Student to be usually smooth. In the case of over-smoothing, the discriminability of hard samples will decrease. We add self-entropy loss in knowledge distillation. Let $y_{mean}^{student}\in R^{k}$ be the average of $\left\{y_{i}^{student}|i=1,\ldots ,N\right\}$, and let $y_{mean}^{student}\left[a\right]$ be the *a*-th element of $y_{mean}^{student}$. Self-entropy loss is(8)$$\begin{array}{c} L_{4}=-\frac{1}{k}\sum_{a=1}^{k}y_{mean}^{student}\left[a\right]log\left(softmax\left(y_{mean}^{student}\left[a\right]\right)\right). \end{array}$$

### 2.4. Interpretability

We can extract knowledge from Teacher and Student, respectively. In Section 2.2, if using knowledge-based mask, the Teacher’s patterns can be directly explained by the gmt files; see Figure 2c. On the contrary, if without the optional knowledge-based mask, Teacher can also automatically learn discriminative patterns. For the interpretation of these patterns, we follow the following steps:Step 1: take the *j*-th linear matrix $W_{j}\in R^{n\times h}$ from Teacher, sum it up to obtain the vector $P_{j}\in R^{n}$, calculate the mean of this vector as $P_{j}^{mean}\in R$, select the indexes with values greater than the mean from $P_{j}$, and use the genes corresponding to these indexes to form a gene set.Step 2: use GSEA (Gene Set Enrichment Analysis) combined with pathway annotation (gmt file on the web or locally) to obtain the pathway name corresponding to this gene set.Step 3: run steps 1 and 2 for other Linear matrices.

In addition, we can extract other information on Teacher. We can treat $\left\{v_{j}\in R^{h}\right\}$ (j=1,…,m) as input and obtain the cell type-specific top-10 important patterns through the difference analysis function in scanpy [9]. For pattern *j*, select the top 2 genes in $P_{j}$ based on their values. Assuming that the indices of these two genes are 1 and 2, we retain $W_{j}\left[1\right]\in R^{h}$ and $W_{j}\left[2\right]\in R^{h}$ as gene embeddings. Finally, we obtained cell type-specific embeddings of 20 genes. Then, we constructed a graph of these important genes based on embedding similarity. We use the graph as a cell type-specific gene co-expression network. Based on the gene co-expression network, we calculate 5 indicators for each gene (degree, betweenness, eigenvalue, pagerank, proximity), and use Q statistics to integrate the 5 indicators to further narrow the scope and obtain cell-type-specific key genes.

For Student, we used SHAP [41] to obtain cell-type-specific genes, which were used to further supplement the key genes obtained from Teacher. For f(z), $z={\left(z_{1},\ldots ,z_{n}\right)}^{⊤}$, $z_{j}$ is *j*-th feature of *z*. We use the Shapley value to evaluate the importance of $z_{j}$ to f(z): (9)$$\begin{array}{c} \phi_{j}\left(z\right)=\\ \sum_{S\subseteq F-\left\{j\right\}}\frac{|S|!(|F|-|S|-1)!}{|F|!}\left(f_{S\cup \left\{j\right\}}\left(z_{S\cup \left\{j\right\}}\right)-f_{S}\left(z_{S}\right)\right), \end{array}$$
where |F| and |S| represent the number of elements contained in *F* and *S*. $F-\left\{j\right\}$ means removing feature *j* from *F*. $z_{F-S}^{\left(i\right)}$ represents the *i*-th observation of a feature not in *S*. $f_{S}\left(z_{S}\right)$ is obtained by averaging $f(z_{S},z_{F-S}^{\left(i\right)})$. The Shapley value measures the importance of the feature by calculating the weighted average of the change in f(z) after removing the *j*-th feature.

For Student $f\left(z\right)=f^{\left(1\right)}⊙f^{\left(2\right)},\ldots ,f^{\left(L\right)}$ (multi-layer dense network), let the output dimension of *l*-th layer be $L^{\left(l\right)}$, the *q*-th input feature of layer *l* is $e_{q}^{\left(l\right)}$, and its mean value is ${\bar{e}}_{q}^{\left(l\right)}$. We obtain the Shapley value through recursion: (10)$$\begin{array}{c} \phi_{q,r}\left(f^{\left(l-1\right)}⊙f^{\left(l\right)},e^{\left(l-1\right)}\right)=\\ \hat{e}\cdot \sum_{r^{\left(l-1\right)}=1}^{L^{\left(l-1\right)}}m^{\left(l-1\right)}\left(r^{\left(l-1\right)},q\right)\cdot m^{\left(l\right)}\left(r,r^{\left(l-1\right)}\right). \end{array}$$(11)$$\begin{array}{c} m^{\left(l-1\right)}\left(r^{\left(l-1\right)},q\right)=\frac{\phi_{q,r^{\left(l-1\right)}}\left(f^{\left(l-1\right)},e^{\left(l-1\right)}\right)}{e_{q}^{\left(l-1\right)}-{\bar{e}}_{q}^{\left(l-1\right)}}. \end{array}$$(12)$$\begin{array}{c} m^{\left(l\right)}\left(r,r^{\left(l-1\right)}\right)=\frac{\phi_{r^{\left(l-1\right)},r}\left(f^{\left(l\right)},e^{\left(l\right)}\right)}{e_{r^{\left(l-1\right)}}^{\left(l\right)}-{\bar{e}}_{r^{\left(l-1\right)}}^{\left(l\right)}}. \end{array}$$

In Equation (10), $\hat{e}=\left(e_{q}^{\left(l-1\right)}-{\bar{e}}_{q}^{\left(l-1\right)}\right)$. Furthermore, $\phi_{q,r}\left(f^{\left(l-1\right)}⊙f^{\left(l\right)},e^{\left(l-1\right)}\right)$ measures $e_{q}^{\left(l-1\right)}$ is relative to the importance of the *r*-th element output by $f^{\left(l-1\right)}⊙f^{\left(l\right)}$. In order to identify genes that are highly correlated with a specific cell type, for each cell, the Shapley value of each input gene relative to the specified cell type needs to be calculated. All Shapley values from cells classified as that type were then taken as absolute values and used to calculate the average. The top 10 genes with the highest average values were interpreted as having high correlations with the cell type.

## 3. Datasets and Experimental Settings

### 3.1. Datasets

For scRNA-seq datasets (RNA modality), there are the following datasets. **BMMC dataset** [44]: scRNA-seq data collected from bone marrow mononuclear cells from 12 healthy human donors, with a total of 13 batches, 22 cell types, and a total cell count of 69,249. **PBMC** [45]: utilizing 10x-Multiome technology to measure 9058 cells of human PBMC. **Pan-cancer dataset** [46]: there are 71,113 myeloid cells in total, and the number of cell types is 23. The dataset contains a total of 13 cancer types, and each cancer type is regarded as a batch. Above datasets are single-cell resolution, and below is the spatial transcriptome dataset. Each sample is a spot (containing multiple cells), and the annotation task is to identify the tissue region type of each spot. **DLPFC dataset** [47]: contains measurement data from 12 batches of adult dorsolateral prefrontal cortex. The dataset contains normalized mRNA expression, 2D spatial coordinates of each spot, and mRNA expression weighted by spatial coordinates (also called niche modal). PLKD only uses mRNA expression and can identify the tissue region to which each spot belongs without the need for other information.

In cell data analysis, researchers also use the ATAC modality to describe a cell (abbreviated as scATAC-seq). Different modalities have different feature spaces, such as accessible chromatin region peaks in ATAC modality and gene expressions in RNA modality. For multi-modal datasets, **10x-Multiome** used 10x-Multiome technology to jointly measure 9631 cells of human PBMC [45]. **Chen-2019** used SNARE-seq technology to jointly measure 9190 cells in mouse cortex [48] (RNA modality: 9190 cells, ATAC modality: 9190 cells). **Ma-2020**: used SHARE-seq technology to jointly measure 32,231 cells in the mouse skin [49]. **Muto-2021** measured 44,190 cells from human kidney using snRNA-seq and snATAC-seq technologies, respectively [50] (RNA modality: 19,985 cells, ATAC modality: 24,205 cells). Statistical information for the datasets is shown in Table 1.

### 3.2. Experimental Settings

Data preprocessing followed the standard settings of the scanpy tutorial. We used scanpy [9] to calculate the size factor. The UMI counts were then divided by the size factor to normalize the counts per cell. Finally, the normalized counts were log1p transformed. For BMMC, we selected the first six batches as the reference dataset and the remaining seven batches as the query dataset. For PBMC, we selected the first batch as the reference dataset and the remaining batch as the query dataset. The Pan-cancer analysis was used to validate the performance of PLKD in cross-disease cell type annotation. We used three batches (ESCA: esophageal carcinoma, THCA: thyroid carcinoma, UCEC: uterine corpus endometrial carcinoma) as the reference dataset, and the remaining diseases (batches) as the query dataset. For DLPFC, we selected the first six slices as the reference dataset and the remaining slices as the query dataset. For the four multi-modal datasets (10x-Multiome, Chen2019 [48], Ma2020 [49], and Muto2021 [50]), we followed the preprocessing pipeline described in CoVEL [51]. Since the lists of highly variable genes vary across datasets, we need to generate dataset-specific pathway masks for each dataset.

We use stochastic gradient descent as the optimizer and adjust the learning rate using cosine learning rate decay. PLKD converges within six epochs. We used the following methods for annotation as baselines: Seurat [8], CellTypist [52], ACTINN [53], TOSICA [17], Cellcano [16], MetaTiME [54], Geneformer [55], scBERT [15], CellLM [56], LangCell [57], scGPT [14], KIDA [18]. We used the following methods for multi-modal integration as baselines: Seurat [8], GLUE [45], harmony [24], LIGER [58], bindSC [59], iNMF [60], CoVEL [51], unioncom [61], scButterfly [62], KIDA [18]. Regarding baseline methods, to ensure a fair comparison, all baseline methods were implemented using their official codebases with default parameters and recommended settings provided in their respective tutorials.

For cell type annotation, we use the following evaluation metrics: Acc (accuracy), Macro F1. For multi-modal integration, we use the following evaluation metric: overall score. The details of the metrics are as follows: **accuracy.** We calculate the accuracy based on the ratio between the total number of cells being classified correctly and the total number of cells. **F1.** For classification results, we calculate true positives (tp), false positives (fp), true negative (tn), and false negatives (fn). F1 score is calculated as follows:(13)$$\begin{array}{c} Precision=\frac{tp}{tp+fp},\\ Recall=\frac{tp}{tp+fn},\\ F1=\frac{2\cdot Precision\cdot Recall}{Precision+Recall}. \end{array}$$

The multi-modal integration metric is **overall score.** We use the same metrics with CoVEL [51]. Biology conservation includes MAP (mean average precision), cell type ASW (cell type average silhouette width), NC (neighbor consistency). Biology conservation is(14)$$\begin{array}{c} biology\;conservation=\frac{MAP+cell\;type\;ASW+NC}{3}. \end{array}$$

Batch removal includes: SAS (Seurat alignment score), omics layer ASW, GC (graph connectivity). Batch removal is(15)$$\begin{array}{c} batch\;remove=\frac{SAS+omics\;layer\;ASW+GC}{3}. \end{array}$$

To compute an overall integration score, we use a 6:4 weight between biology conservation and batch removal:(16)$$\begin{array}{c} overall\;score=0.4\times batch\;remove+0.6\times biology\;conservation. \end{array}$$

## 4. Results

### 4.1. Cell Type Annotation

We evaluated the performance of PLKD in cell type annotation tasks on multiple standard benchmark datasets, including PBMC, BMMC, DLPFC, and Pan-cancer, to verify the model’s ability to accurately predict cell identities across different biological scenarios. Quantitative results are presented in Table 2, and qualitative visualization results are shown in Figure 3 and Figure 4.

Table 2 shows the comparative results between PLKD and representative baseline methods, including Seurat, ACTINN, CellTypist, scBERT, TOSICA, and KIDA. Across all datasets, PLKD achieved the highest or comparable accuracy and macro-averaged F1 score. Notably, in datasets with strong batch effects (e.g., Pan-cancer) or spatial heterogeneity (e.g., DLPFC), PLKD still maintained robust annotation performance. These results indicate that PLKD can effectively integrate biological patterns and transcriptional signals to provide consistent and reliable cell type annotation.

Figure 3 presents the visualization results of cell embeddings from the BMMC and Pan-cancer datasets. In both cases, PLKD exhibited clearer cell type boundaries compared to the original PCA embeddings. Cells of the same type clustered closely in the low-dimensional space, with distinct separation between different types. This demonstrates that the representations learned by PLKD can capture biological structures associated with functional differences. Additionally, when compared to the PCA embedding of raw counts (colored by batch labels in the first column of the figure), PLKD eliminated batch effects and aligned samples with different batch labels but the same cell type labels.

To quantitatively evaluate the batch integration performance, we calculated kBET (k-nearest neighbor batch effect test), and ASW (average silhouette width for cell type labels). As shown in Table 3, PLKD significantly outperforms the baseline PCA embedding on both datasets. These quantitative results, consistent with the visual inspections in Figure 3, confirm that PLKD effectively removes batch effects while preserving biological heterogeneity.

Figure 4 shows the annotation results of spatial transcriptomic sections from the DLPFC dataset. The first row displays the true tissue annotations, and the second row shows the prediction results of PLKD. It can be observed that the tissue structure predicted by PLKD was highly consistent with the true spatial distribution, with good continuity in identifying various tissue regions and only a small number of misclassified samples. Notably, PLKD achieved accurate recognition results at boundaries, enabling clear separation of different tissue regions. This indicates that PLKD can accurately transfer cell identity information to the spatial domain while maintaining spatial coherence and biological authenticity.

In addition to superior annotation accuracy, PLKD demonstrates significant computational efficiency during the inference phase, validating the effectiveness of our knowledge distillation strategy. The Teacher model, designed to learn complex patterns, contains a total of 28,874,173 parameters. In contrast, the Student model, which is deployed for actual inference, is significantly more compact with only 547,597 parameters. This represents a reduction in model size to merely 1.9% of the Teacher model. This lightweight architecture ensures that PLKD can perform rapid and efficient inference on large-scale datasets without compromising the robust performance inherited from the Teacher.

Overall, PLKD demonstrated strong performance and generalization ability across different types of single-cell data. Its comprehensive advantages in accurate classification, biologically interpretable embeddings, and spatial consistency support PLKD as a unified framework, which can be widely applied to cell type annotation tasks for scRNA-seq and spatial transcriptomic data.

### 4.2. Gene Inference

To ensure the reliability and interpretability of the identified marker genes, we employed a dual-validation strategy. We defined the candidate gene pool as the intersection of genes identified by the Teacher model (via pattern learning) and the Student model (via SHAP inference). This consensus approach ensures that the selected genes are supported by both global pattern structures and local decision boundaries.

We performed gene inference across different cell types to further understand the performance of the PLKD model in gene expression profile analysis. By analyzing the gene expression of cell types such as CD8+ T cells, erythroblasts, and plasma cells, we inferred key genes associated with the characteristics of these cell types.

As shown in Figure 5, while the intersection sets contain both cell-type-specific markers and some common housekeeping genes, the SHAP value analysis from the Student model effectively ranks the biologically relevant markers at the top.

For CD8+ T cells, the inferred genes include LYST, FUT8, MYO1E, IL7R, HLA-DRA, SEL1L3, PHEX, RGS2, and TNFAIP3. These genes play important roles in immune response, signal transduction, and antigen recognition of CD8+ T cells. Specifically, IL7R has a critical function in T cell development, HLA-DRA is involved in antigen presentation, and TNFAIP3 exerts its role in intracellular inflammatory signal transduction.

For erythroblasts, we inferred that two genes, AHSP and HBB, are key markers. HBB is a core gene encoding hemoglobin and plays a vital role in erythrocyte development, while AHSP plays an important part in iron metabolism and hemoglobin stability of erythrocytes.

For plasma cells, the inferred key genes are MZB1 and IGKC. MZB1 is involved in the functionalization and antibody secretion of plasma cells, and IGKC is a component of immunoglobulin heavy chains, marking the active state of plasma cells in the immune response.

These inference results demonstrate the effectiveness of PLKD in identifying and analyzing cell-specific signature genes, further verifying its broad application potential in biological tasks.

### 4.3. Cross-Modal Annotation

PLKD can be used for ATAC modality annotation. For the ATAC modality, to align the feature space with RNA data, we calculated gene activity scores (GAS) using seurat [8]. This step aggregates reads within gene bodies and promoter regions to quantify gene-level accessibility, enabling the application of gene-based pathway masks across both modalities.

The performance of PLKD in cross-modal annotation tasks demonstrates its strong versatility and consistency. Through the annotation of RNA modality and ATAC modality, we can verify the model’s excellent adaptability and high accuracy across different data types. To evaluate PLKD’s performance under these different modalities, we used RNA and ATAC data from the same subject and compared the annotation results of the two modalities.

As shown in Table 4, for data from the same subject, PLKD achieves high accuracy in annotation results for both RNA and ATAC modalities, with a high degree of consistency between the two. Specifically, whether for the annotation of gene expression profiles or chromatin accessibility regions, PLKD can identify similar cell types and their biological characteristics under both modalities. This indicates that PLKD can not only perform effective annotation in a single modality, but also maintain the accuracy and consistency of annotation in a cross-modal context.

This cross-modal annotation capability is of great significance for the integration of multi-omics data, as it provides a solid foundation for subsequent multi-modal analysis. In complex biological research, obtaining consistent results from RNA and ATAC modalities will greatly improve the credibility of data interpretation, and provide a more comprehensive perspective for studies on disease mechanisms, gene regulatory networks, and other related fields.

### 4.4. Ablation Study

Ablation study on 10x-Multiome [45], where we evaluated cell type annotation with Acc, and evaluated batch integration with ARI. For standard settings (using divergence-based clustering loss and self-entropy loss, using optional knowledge-based mask, and not adding noise to reference labels), both Teacher and Student can achieve good performance, see the first row of Table 5. “(T)” means “Teacher”, “(S)” means “Student”. Compared with the standard situation, we set up three situations in ablation study:
(1)Neither divergence-based clustering loss nor self-entropy loss is used, the optional knowledge-based mask is used, and no noise is added to reference labels; see row 2 of Table 5.(2)Use divergence-based clustering loss and self-entropy loss, do not use the optional knowledge-based mask, and do not add noise to reference labels; see row 3 of Table 5.(3)Use divergence-based clustering loss and self-entropy loss, use optional knowledge-based mask, and add noise to reference labels. For example, randomly modify the original correct label to another label, and the noise ratio is 0.1, 0.15 and 0.2, respectively; see rows 4–6 of Table 5.

For the first situation, both Teacher and Student’s cell type annotation and batch correction performance suffered a serious decline. Since divergence-based clustering loss and self-entropy loss are designed for batch integration, we can see a decrease in batch integration performance. Note that the annotation accuracy is also declining, which verifies the relationship between batch integration and annotation tasks. For the second situation, although the performance of Teacher and Student has declined, it is not obvious. This shows that PLKD can learn discriminative patterns from reference dataset without a clear initialization scheme. For the third situation, when noise is gradually added to reference labels, the performance of both Teacher and Student decreases, but Student decreases more slowly than Teacher. This is because distillation learning allows Student to tolerate a certain amount of noise.

## 5. Discussion

In cross-modal annotation, we utilized gene activity scoring to convert the ATAC modality into the RNA feature space. Consequently, we were able to derive embeddings for the multi-modal data. For multi-modal integration, we simply concatenated these embeddings directly.

Figure 6 presents the multi-modal integration results across different datasets, specifically comparing the performance of our proposed PLKD method with several existing mainstream methods. To further understand the advantages of our method in multi-modal integration tasks, we conducted evaluations using multiple classic datasets, including the 10x-Multiome, Chen2019 [48], Ma2020 [49], and Muto2021 [50] datasets. The evaluation metric for each dataset is the overall score, where a higher score indicates better performance of the method.

The bar chart in Figure 6 compares the performance of the PLKD method with other multi-modal integration methods—including KIDA, CoVEL, scButterfly, GLUE, Seurat, Harmony, LIGER, bindSC, iNMF, and unioncom—across these datasets. It can be seen that PLKD achieves higher scores than other methods on all datasets, with outstanding performance particularly on the 10x-Multiome dataset, reaching a score of 0.889, which is significantly higher than that of other methods. In contrast, the KIDA method, which performs most closely to PLKD, achieves a score of 0.856 on the same dataset, showing a clear gap.

PLKD also performs excellently on other datasets. For example, in the Chen2019 dataset, PLKD achieves a score of 0.835, which is more than 10% higher than that of other methods, especially CoVEL and scButterfly. On the Muto2021 dataset, PLKD also demonstrates strong performance with a score of 0.866, outperforming most methods.

In biological datasets, rare or minor cell populations often carry significant biological insights but are difficult to identify due to class imbalance. To assess PLKD’s robustness in this scenario, we performed a stratified evaluation, grouping the Pan-cancer dataset by abundance: high (>5000 cells), medium (1000–5000 cells), and low (<1000 cells). As shown in Table 6, PLKD demonstrates remarkable stability across all groups. Notably, for the Low abundance group (containing cell types with as few as 171 cells), PLKD achieved a perfect Recall score of 1.000 and a high F1-score of 0.848. This indicates that PLKD is not biased towards dominant classes and effectively captures the discriminative patterns of minor populations.

To further rigorously assess the limits of the model under extreme class imbalance, we conducted a subsampling stress test. We paired the major cell type (*Mono_CD14*, N = 11,511) against three minor cell types of varying difficulty and simulated imbalance ratios ranging from 100:1 to 500:1. As illustrated in Figure 7, PLKD exhibits a scientifically realistic performance trend: while F1-scores are robust (≈0.76 for *cDC2_CXCL9*) at a 100:1 ratio, they gradually decline as the ratio approaches 500:1 (where minor cells account for <0.2% of the population). This degradation is statistically expected, as the impact of background false positives on precision becomes magnified when the signal is extremely scarce. However, crucially, the model performance does not collapse (e.g., F1 remains ≈0.46 for *cDC2_CXCL9* at 500:1), confirming that PLKD relies on learned molecular patterns rather than class priors to identify rare populations.

## 6. Conclusions

In this study, we addressed the critical challenge of domain gaps caused by batch effects in single-cell data analysis, and proposed PLKD—a cell type annotation method integrating pattern learning and knowledge distillation—to enhance the accuracy, robustness, and versatility of multi-scenario single-cell data processing.

PLKD’s dual-model architecture (Teacher-Transformer and Student-MLP) delivers distinct advantages. The Teacher model groups genes into biologically meaningful patterns via self-attention, shifting focus from batch-sensitive gene-level expression to function-level interactions. This design, combined with divergence-based clustering loss and cross-entropy loss, effectively eliminates batch effects while enhancing the discriminability of cell representations. As validated on benchmark datasets (BMMC, PBMC, DLPFC, Pan-cancer), PLKD achieved the highest or comparable Accuracy and Macro F1-scores among state-of-the-art methods. Even in datasets with strong batch effects (Pan-cancer) or spatial heterogeneity (DLPFC), it maintained robust performance, as visualized by clearer cell type boundaries in UMAP embeddings and high consistency with true tissue distributions in spatial transcriptomic annotation.

Beyond single-modal cell type annotation, PLKD exhibits strong cross-modal and multi-modal capabilities. In cross-modal annotation (Table 4), it achieved high and consistent accuracy for both RNA and ATAC modalities from the same subject, effectively capturing biological correlations between different molecular layers. In multi-modal integration (Figure 6), PLKD outperformed 10 mainstream methods on all datasets, with an Overall Score of 0.889 on 10x-Multiome (surpassing the closest competitor KIDA by 0.033) and over 10% improvement compared to CoVEL and scButterfly on Chen2019, demonstrating its superiority in fusing multi-modal data.

Gene inference results further validated PLKD’s biological interpretability. By extracting key genes for specific cell types (e.g., IL7R and TNFAIP3 for CD8+ T cells, HBB and AHSP for erythroblasts, MZB1 and IGKC for plasma cells), it links model outputs to functional biological mechanisms, providing valuable insights for downstream research. Ablation studies (Table 5) confirmed that divergence-based clustering loss and self-entropy loss are critical for maintaining performance—their removal led to a significant drop in Accuracy and ARI. Additionally, the Student model showed greater noise tolerance than the Teacher model, attributed to knowledge distillation mitigating label noise.

In summary, PLKD integrates pattern learning, knowledge distillation, and multi-task optimization to address core challenges in single-cell data analysis, including batch effects, multi-modal integration, and biological interpretability. Its lightweight Student architecture enables efficient annotation of large-scale query datasets, while its cross-modal and multi-modal capabilities expand its applicability to diverse omics data. As a unified framework, PLKD holds broad potential for advancing research in disease mechanisms, gene regulatory networks, and spatial transcriptomics, providing a reliable tool for bioinformatics researchers.

Despite the promising results, PLKD has limitations that warrant further investigation. First, like most supervised methods, PLKD’s performance relies on the quality and comprehensiveness of the reference atlas. If the reference dataset contains misannotations or lacks specific cell states, the Student model may propagate these biases. Second, while PLKD can handle batch effects robustly, identifying “novel” cell types (those present in the query but absent in the reference) remains a challenge. Currently, these cells might be forced into existing categories with low confidence. In future work, we plan to incorporate uncertainty estimation and open-set recognition mechanisms to flag unknown cell populations, further enhancing the tool’s utility for discovery in large-scale single-cell analysis.

## Figures and Tables

**Figure 1 biology-15-00002-f001:**
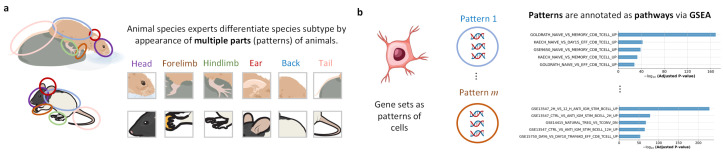
Demonstration of using patterns as discriminative features. (**a**) Illustration of patterns by image classification. (**b**) Transferring image classification to cell identification, gene sets in scRNA-seq are treated as patterns.

**Figure 2 biology-15-00002-f002:**
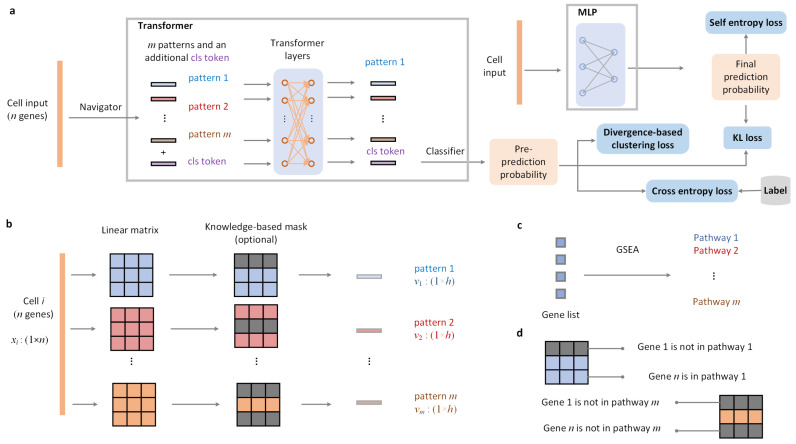
The overall of PLKD. (**a**) PLKD contains Teacher (Transformer) and Student (MLP). Navigator module converts a cell from input gene expressions into *m* patterns. (**b**) Navigator module transforms the input gene expression vector into pattern tokens using parallel linear matrices. (**c**) Knowledge-based mask: obtaining pathways from gene list. (**d**) Knowledge-based mask: add the mask based on the relationship between pathways and genes.

**Figure 3 biology-15-00002-f003:**
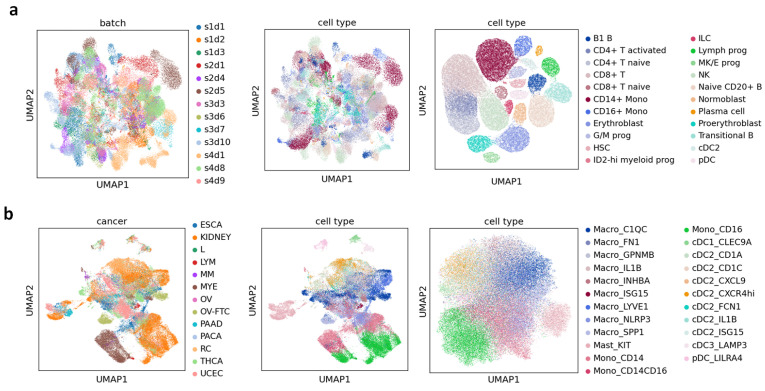
UMAP visualization of cell embeddings. (**a**) UMAP on BMMC dataset, left to right: raw data PCA embedding (batch label coloring), raw data PCA embedding (cell type label coloring), PLKD embedding (cell type label coloring). (**b**) UMAP on Pan-cancer dataset, left to right: raw data PCA embedding (cancer/batch label coloring), raw data PCA embedding (cell type label coloring), PLKD embedding (cell type label coloring).

**Figure 4 biology-15-00002-f004:**
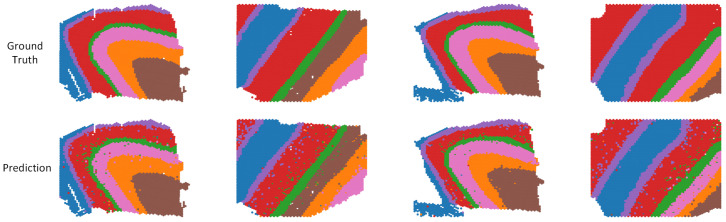
Visualization on slices, we compare the actual cell types with the predicted cell types. Different colors represent different cell types.

**Figure 5 biology-15-00002-f005:**
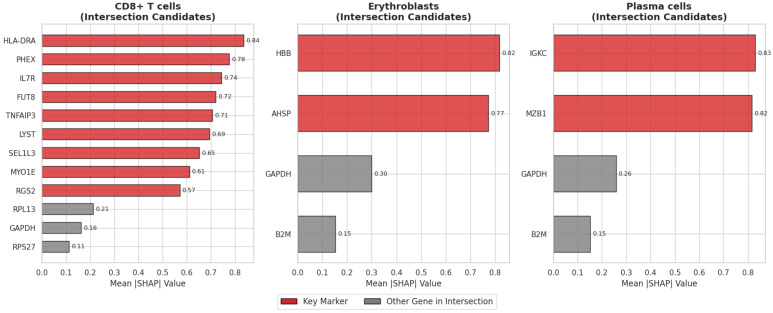
Quantitative ranking of intersection genes by SHAP value.

**Figure 6 biology-15-00002-f006:**
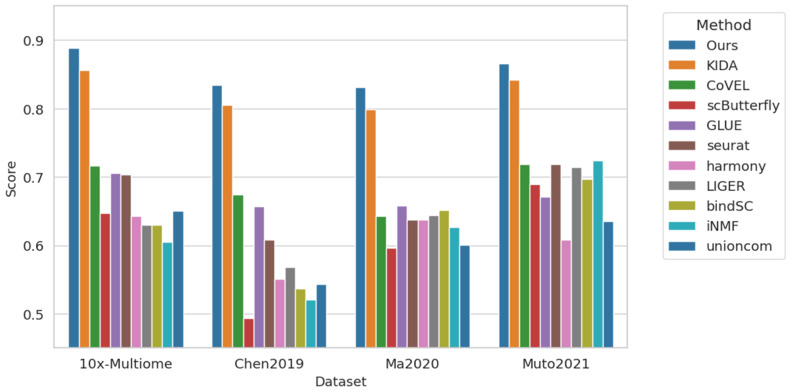
Multi-modal integration results on different datasets [48,49,50].

**Figure 7 biology-15-00002-f007:**
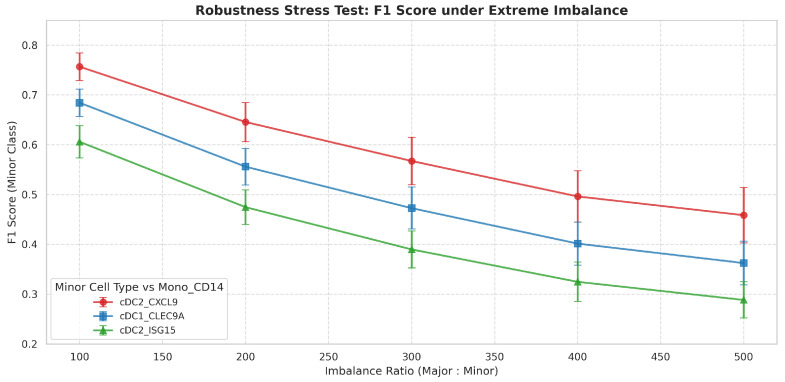
Robustness evaluation under extreme class imbalance. The line chart illustrates the F1-score degradation of three minor cell types (*cDC2_CXCL9*, *cDC1_CLEC9A*, *cDC2_ISG15*) as their ratio to the major cell type (*Mono_CD14*) increases from 100:1 to 500:1, while performance naturally declines due to sparsity, and PLKD maintains discriminative ability even at the 500:1 ratio.

**Table 1 biology-15-00002-t001:** The statistics of datasets.

DatasetName	Task	Cell Counts	Type Counts	Batch Counts	Modality	Organ
BMMC	annotation	69,249	22	13	RNA	HomoBMMC
PBMC	annotation	9058	19	2	RNA	HomoPBMC
Pan-cancer	annotation	71,113	23	13	RNA	HomoCancers
DLPFC	annotation	47,329	7	12	Spatial RNA	HomoCortex
10x-Multiome	integration	9631 × 2	19	N/A	RNA + ATAC	HomoPBMC
Chen-2019 [48]	integration	9190 × 2	22	N/A	RNA + ATAC	MouseCortex
Ma-2020 [49]	integration	32,231	22	4	RNA + ATAC	MouseSkin
Muto-2021 [50]	integration	44,190	13	5	RNA + ATAC	HomoKidney

**Table 2 biology-15-00002-t002:** Annotation results on cell type classification benchmarks.

	BMMC		PBMC		DLPFC		Pan-Cancer	
**Method**	**Acc**	**F1**	**Acc**	**F1**	**Acc**	**F1**	**Acc**	**F1**
Seurat	0.714	0.528	0.688	0.605	0.646	0.630	0.758	0.737
ACTINN	0.826	0.803	0.874	0.800	0.729	0.704	0.787	0.760
CellTypist	0.866	0.852	0.931	0.930	0.787	0.702	0.712	0.649
TOSICA	0.912	0.902	0.958	0.935	0.939	0.931	0.885	0.863
Cellcano	0.877	0.826	0.958	0.955	0.838	0.761	0.748	0.716
MetaTiME	0.910	0.913	0.870	0.855	0.738	0.731	0.869	0.842
Geneformer	0.931	0.897	0.968	0.966	0.947	0.927	0.856	0.846
scBERT	0.906	0.885	0.979	0.978	0.930	0.897	0.870	0.832
CellLM	0.922	0.896	0.966	0.963	0.934	0.930	0.836	0.821
LangCell	0.930	0.896	0.980	0.958	0.949	0.932	0.874	0.834
scGPT	0.925	0.897	0.945	0.951	0.953	0.927	0.866	0.849
KIDA	0.931	0.902	0.980	0.972	0.953	0.947	0.891	0.866
**Ours**	**0.944**	**0.921**	**0.971**	**0.963**	**0.961**	**0.955**	**0.916**	**0.902**

**Table 3 biology-15-00002-t003:** Evaluation of batch integration performance on BMMC and Pan-cancer datasets. Bold text indicates the best results.

Dataset	Method	kBET	ASW (Cell Type)
BMMC	PCA	0.32	0.28
	PLKD	**0.89**	**0.81**
Pan-cancer	PCA	0.15	0.35
	PLKD	**0.78**	**0.64**

**Table 4 biology-15-00002-t004:** Cross-modal annotation results. Bold text indicates the best results.

	Ma2020 (RNA)		Ma2020 (ATAC)		Muto2021 (RNA)		Muto2021 (ATAC)	
	**[49]**		**[49]**		**[50]**		**[50]**	
**Method**	**Acc**	**F1**	**Acc**	**F1**	**Acc**	**F1**	**Acc**	**F1**
Seurat	0.912	**0.905**	0.701	0.700	0.748	0.739	0.869	0.864
Actinn	0.854	0.839	0.791	0.777	0.812	0.799	0.760	0.752
CellTypist	0.697	0.643	0.668	0.612	0.755	0.714	0.723	0.701
Tosica	**0.915**	0.903	0.905	0.905	**0.874**	**0.869**	0.821	0.819
Cellcano	0.864	0.853	**0.946**	**0.940**	0.855	0.841	**0.922**	**0.921**
MetaTiME	0.645	0.613	0.601	0.601	0.656	0.649	0.603	0.602
Geneformer	0.929	0.916	0.906	0.897	0.888	0.861	0.882	**0.871**
scBERT	0.933	**0.928**	**0.922**	**0.920**	0.894	0.890	0.868	0.867
CellLM	0.941	0.917	0.909	0.883	0.908	0.853	0.877	0.841
LangCell	**0.948**	0.922	0.916	0.896	**0.921**	**0.898**	0.842	0.829
scGPT	0.940	0.926	0.914	0.909	0.901	0.894	**0.879**	0.862
KIDA	0.949	0.932	0.951	0.944	0.925	0.917	0.923	0.914
Ours	**0.951**	**0.939**	**0.956**	**0.952**	**0.938**	**0.922**	**0.936**	**0.921**

**Table 5 biology-15-00002-t005:** Ablation experiment results: ‘loss’ represents whether to use divergence-based clustering loss and self-entropy loss; ‘mask’ represents whether to use the optional knowledge-based mask; and ‘noise’ represents the ratio of noise added to reference labels. T: Teacher, S: Student.

Loss	Mask	Noise	Acc (T)	ARI (T)	Acc (S)	ARI (S)
✓	✓	0	0.974	0.935	0.981	0.966
×	✓	0	0.905	0.526	0.897	0.405
✓	×	0	0.962	0.928	0.969	0.914
✓	✓	0.1	0.942	0.908	0.944	0.906
✓	✓	0.15	0.892	0.868	0.901	0.831
✓	✓	0.2	0.868	0.768	0.862	0.795

**Table 6 biology-15-00002-t006:** Performance stratified by cell type abundance on the Pan-cancer dataset. PLKD maintains high performance for rare cell types (low group).

Group	Cell Count Range	F1-Score	Recall	Precision
High	>5000	0.831	0.774	0.982
Medium	1000–5000	0.888	0.997	0.824
Low	<1000	0.848	1.000	0.771

## Data Availability

Data and Code is available at https://github.com/mz-hznu-0408/PLKD, accessed on 15 December 2025, the code references TOSICA (https://github.com/JackieHanLab/TOSICA, accessed on 15 December 2025). The datasets used in this study are already published and were obtained from public data repositories.

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
