# Peer review of "Pattern Learning and Knowledge Distillation for Single-Cell Data Annotation"

_biology, 2025, doi:10.3390/biology15010002_

Round 1
Reviewer 1 Report
Comments and Suggestions for Authors
The authors propose a PLKD, a supervised single-cell annotation framework combining pattern learning (via a Transformer “Teacher”) and knowledge distillation (into an MLP “Student”). The key idea is that grouping genes into “patterns” (with optional pathway-based masking) yields representations less sensitive to batch effects, while the Teacher-Student architecture enables efficient downstream annotation. While the combination of ideas is somewhat new, several core components resemble existing methods: TOSICA already incorporates pathway-level representations using Transformers. KIDA uses knowledge distillation for robust annotation.
I have several comments below:
- No source code link or reproducible implementation is provided. I am not fully confident about the method’s performance without independently running PLKD. Please provide a minimal runnable demo or notebook showing how PLKD performs cell type annotation on a small example dataset.
- The Materials and Methods section heavily focuses on describing PLKD, but does not provide sufficient details on the experimental setup for other baseline methods. It is unclear whether all baselines were run with default parameters, author-recommended settings, or dataset-specific tuning.
- Please provide details about preprocessing for each dataset, if applicable.
- I am not sure if the same pathway mask is applied across all datasets or generated separately for each dataset. Could you clarify it?
- The manuscript claims lightweight inference, but does not report the number of parameters in the Teacher and Student models. Please provide a parameter count and/or model size for both models.
- The method focuses on cell annotation but claims it can improve batch correction. However, batch-effect metrics such as LISI, kBET, silhouette width, or entropy mixing scores are not reported, making it hard to assess whether batch integration is genuinely improved rather than merely visually appealing (Figure 3).
- In multi-modal experiments, please explain how RNA and ATAC features are aligned. For example, was gene activity scoring applied?
- In many biological datasets, rare or minor cell populations are particularly important. Please comment on whether PLKD can robustly annotate minor cell types, and if possible, include an evaluation of performance stratified by cell type abundance.
Reviewer 2 Report
Comments and Suggestions for Authors
The manuscript by Zhang, Ren and Li presents PLKD, a pattern learning based method for single-cell multi-modal data integration and cell type annotation method. This method employs the knowledge distillation technique in a dual-model network architecture composed of a Teacher model (Transformer) and a light-weight Student model (MLP). By pathway-level pattern extraction in Teacher and transferring the knowledge from Teacher to Student, PLKD demonstrated capability of removing batch effects and identifying clusters out of single-cell heterogeneity. Moreover, by analysis of the Shapley value in Student, the authors showed that PLKD was able to predict key marker genes for specific cell types.
Comments and suggestions:
- Line 52-54, “For batch effects removal, popular methods such as Seurat [8,20] and MNN [21] rely on mutual nearest neighbor methods to eliminate batch effects.” Seurat is a package that wraps up a host of batch correction algorithms, including CCA, MNN and Harmony. Although some literature did list “Seurat” and “MNN” side by side, it would be more precise to write out the underlying algorithm(s) instead of “Seurat”.
- Figure 2b. The caption should be more detailed.
- Line 137. The line overran and got the formula truncated.
- Line 138. What does 𝑟 represent in 𝑐𝑟? (“reference”?)
- Line 156-157. Similar question as the previous remark for 𝑐𝑟,𝑖 and 𝑐𝑟,𝑗.
- Line 158-161. What is 𝑆(⋅) as in 𝑆(𝑌teacher𝑏)? Is it not a similarity matrix (Line 156)
- Line 197, “For pattern j, select the top-2 genes in Pj based on their values”. Why two genes?
- Line 227, “Use scanpy to calculate the size factor.” Please add version number for scanpy as well as any other software packages used in this work.
- Line 244-251, Table 1. It would be better if the authors could add the number of cell types as an extra column to the table.
- Line 252, “…, we split batch groups of the dataset into the validation set and the rest for training.” At what ratio?
- Line 256, “For multi-modal integration, we use the following evaluation metric: Overall score.” What is the “overall score”? It seems that its definition cannot be found.
- Table 2, “Annotation results on cell type classification benchmarks.”. Since one major highlight of PLKD is that it can transfer knowledge from reference to query data, it would be critical to document quantitatively how well it can perform.
- Line 298-317, Section 3.2 (“Gene inference”) can be improved in two aspects. First, according to Methods, the “interpretability genes” can be inferred from both Teacher and Student, but here it is unclear which came result from which method. Second, the whole section is overwhelmed by literacy but lacks of quantitative exhibition. For example, if the authors prioritized the genes by their SHAP values, how much would they align with the importance from domain knowledge. One
or two figures similar to what UnitedNet Supplementary Figure 9 showed would suffice. - Line 425, “Data availability”. The authors should include the source and/or download link for each dataset analyzed in this study.
- Although not mandatory the authors are recommended to make their software and benchmark publicly available. That will be great for not only the interest of the community but also the sake of research reproducibility.
Reviewer 3 Report
Comments and Suggestions for Authors
Zhang et al have presented a detailed and well-written manuscript where they presented pattern learning and knowledge distillation for single-cell data annotation.
The manuscript is very promising, and I would suggest authors address a list of major comments before this paper can be accepted
Major
- Authors should provide the scope and application of PLKD and what’s its place will be in currently existing single cell pipelines, and how it stands compared to a currently existed label transfer and deep learning annotators
- Authors should define the integration metrics clearly, provide batch vs biology metrics across benchmarks and provide a possible comparison of how PPLKD embeddings performed compared to the Harmony + Seurat sc analysis
- Please provide details regarding multimodal design. How do authors address the ATAC seq integration with RNA dataset. Please provide the benchmarks of multi-mic integration and provide detailed cross-modal evaluation and multimodal methods in the method section
- Please provide how the loss formulation was performed, how the weights were chosen and tuned. Authors should describe the complete Teacher and Student losses. Authors should also discuss the training stability and convergence properties.
Minor
• ⁃ Please add the limitation paragraph to the discussion and conclusion section, acknowledging the current limitations of the PLKD method and possible challenges of its implementation in future sc analysis, and how the authors think they can overcome them
Round 2
Reviewer 1 Report
Comments and Suggestions for Authors
I could not locate the demo data from the PLKD repository (https://github.com/mz-hznu-0408/PLKD), which prevents me from fully testing the package.
In addition, after examining the code in detail, I noticed that the implementation appears to rely heavily on TOSICA (“Transformer for One Stop Interpretable Cell Type Annotation,” Nat. Commun. 2023). Many components, including core functions, scripts, and the batch integration concept, are identical or nearly identical to TOSICA. Aside from adding a simple Student MLP module, the model architecture and methodological framework remain unchanged. Based on the current code and description, the novelty of the work is extremely limited, and the degree of overlap raises concerns about originality and potential plagiarism. Importantly, the manuscript does not clearly disclose that the model is built directly upon TOSICA. The brief statement “We are inspired by the pathway-based method [17]” is insufficient and does not adequately acknowledge the degree to which TOSICA forms the foundation of the method.
For example, the only difference in the model architecture is the addition of the Student module
https://github.com/mz-hznu-0408/PLKD/blob/main/PLKD/PLKD_model.py
https://github.com/JackieHanLab/TOSICA/blob/main/TOSICA/TOSICA_model.py
```
admin@local $ diff PLKD_model.py ../../TOSICA/TOSICA/TOSICA_model.py
257,278d256
< class Student(nn.Module):
< def __init__(self, input_dim, num_classes, hidden_dims=[256, 128], dropout=0.1):
< super().__init__()
< layers = []
< prev_dim = input_dim
< for dim in hidden_dims:
< layers.append(nn.Linear(prev_dim, dim))
< layers.append(nn.LayerNorm(dim))
< layers.append(nn.GELU())
< layers.append(nn.Dropout(dropout))
< prev_dim = dim
< self.encoder = nn.Sequential(*layers)
< self.head = nn.Linear(prev_dim, num_classes)
<
< def forward(self, x):
< latent = self.encoder(x)
< logits = self.head(latent)
< return logits
<
< def get_latent(self, x):
< return self.encoder(x)
<
```
Regarding the minor cell type issue raised in the previous review round, categorizing cell types into three abundance-based groups (High, Medium, Low) does not adequately address how many genuinely minor cell types are present in the datasets. As currently designed, this experiment cannot directly evaluate model performance on true minor cell types.
A more appropriate approach would be to quantify the ratio between major and minor cell types and explicitly assess performance across a range of imbalance scenarios. For example, one could subsample from the pan-cancer dataset to generate extreme major–minor ratios and evaluate whether the model maintains robustness under conditions where minor cell types are very scarce.
Reviewer 2 Report
Comments and Suggestions for Authors
The authors have addressed my concerns in the revision. I do not have more questions.
Author Response
Dear Reviewer, Thank you very much for your careful review and valuable feedback on our manuscript. We greatly appreciate that you have recognized our revisions addressing your concerns, and we are grateful for your time and efforts dedicated to improving our work. We wish the manuscript could proceed smoothly in the subsequent process, and thank you again for your support! Best regards,Reviewer 3 Report
Comments and Suggestions for Authors
Zhang et al have addressed my concerns. After rereading the manuscript in its current form, I would suggest one minor comment: add the section on how to use their pipeline and how it can be implemented in the computational biology and non-computational biology labs. This should improve the visibility of their algorithm and increase their future citations.
Otherwise, i would recommend to accept this manuscript in its current form
Author Response
Dear Reviewer, Thank you very much for your careful review and valuable feedback on our manuscript. We greatly appreciate that you have recognized our revisions addressing your concerns, and we are grateful for your time and efforts dedicated to improving our work. We wish the manuscript could proceed smoothly in the subsequent process, and thank you again for your support! Best regards,Round 3
Reviewer 1 Report
Comments and Suggestions for Authors
All of my comments have been satisfactorily addressed in the revised manuscript. I was able to run the demo code successfully. I believe the manuscript is now ready for publication.